# Isolation of *Artemisia capillaris* membrane-bound di-prenyltransferase for phenylpropanoids and redesign of artepillin C in yeast

Ryosuke Munakata[1,2], Tomoya Takemura[1], Kanade Tatsumi[1], Eiko Moriyoshi [1], Koki Yanagihara[1], Akifumi Sugiyama[1], Hideyuki Suzuki [3], Hikaru Seki[4], Toshiya Muranaka[4], Noriaki Kawano[5], Kayo Yoshimatsu[5], Nobuo Kawahara[5], Takao Yamaura[6], Jérémy Grosjean[2], Frédéric Bourgaud[7], Alain Hehn [2] & Kazufumi Yazaki[1]*

Plants produce various prenylated phenolic metabolites, including flavonoids, phloroglucinols, and coumarins, many of which have multiple prenyl moieties and display various biological activities. Prenylated phenylpropanes, such as artepillin C (3,5-diprenyl-*p*-coumaric acid), exhibit a broad range of pharmaceutical effects. To date, however, no prenyltransferases (PTs) involved in the biosynthesis of phenylpropanes and no plant enzymes that introduce multiple prenyl residues to native substrates with different regio-specificities have been identified. This study describes the isolation from *Artemisia capillaris* of a phenylpropane-specific PT gene, *AcPT1*, belonging to UbiA superfamily. This gene encodes a membrane-bound enzyme, which accepts *p*-coumaric acid as its specific substrate and transfers two prenyl residues stepwise to yield artepillin C. These findings provide novel insights into the molecular evolution of this gene family, contributing to the chemical diversification of plant specialized metabolites. These results also enabled the design of a yeast platform for the synthetic biology of artepillin C.

[1] Laboratory of Plant Gene Expression, Research Institute for Sustainable Humanosphere, Kyoto University, Uji, Kyoto 611–0011, Japan. [2] Université de Lorraine, INRA, LAE, F54000 Nancy, France. [3] Department of Research & Development, Kazusa DNA Research Institute, Kisarazu 292-0818, Japan. [4] Department of Biotechnology, Graduate School of Engineering, Osaka University, Suita 565-0871, Japan. [5] Tsukuba Division, Research Center for Medicinal Plant Resources, National Institutes of Biomedical Innovation, Health and Nutrition, Tsukuba 305-0843, Japan. [6] The Yamashina Botanical Research Institute, Nippon Shinyaku Co. Ltd., 39 Sakanotsuji-cho, Ohyake, Yamashina-ku, Kyoto 607-8182, Japan. [7] Plant Advanced Technologies – PAT, 19 Avenue de la forêt de Haye, 54500 Vandoeuvre, France. *email: yazaki@rish.kyoto-u.ac.jp

Prenylated polyphenolic compounds are a class of composite-type plant specialized metabolites, which are biosynthesized by the coupling of two major metabolic pathways in plants, the isoprenoid and shikimate/polyketide pathways[1]. These hybrid-type metabolites have been identified as the bioactive components in many medicinal plants; to date, about 1000 of these compounds have been chemically and biologically characterized[1]. The physiological and pharmacological effects of these compounds differ widely, with many having antioxidant, antibacterial, antitumor, and estrogenic activities, with some having anti-obesity and anti-muscle atrophy properties[2–4]. These specialized metabolites can be classified according to the structures of their core nuclei. The largest group consists of flavonoids, followed by coumarins, phloroglucinols, xanthones, stilbenes, and simple phenylpropanoids. In particular, the di-prenylated p-coumaric acid, artepillin C has been extensively studied over the last two decades due to its close relationship to propolis, a valuable apicultural product.

Propolis is a resinous substance prepared by honeybees, which collect resins from buds and other aerial parts of different plant species, and used by honeybees to seal physical defects (e.g., cracks and holes) in beehives, to prevent invasion by their predators[5]. This honeybee product has been sold worldwide in natural medicines and food supplements because of its broad pharmaceutical and health-promoting activities, which have been attributed to its complex chemistry involving over 500 constituents[5,6]. The chemical composition of propolis is highly dependent on its botanical sources, geological locations, and bee species, with each type of propolis having a unique spectrum of bioactivities[5]. Brazilian green propolis is one of the most globally widespread types used for commercial purposes and is characterized by the presence of bioactive prenylated derivatives of p-coumaric acid, such as drupanin and artepillin C, as major constituents, with concentrations frequently exceeding 10%[6–9]. These ingredients were exploited as collation markers to identify the botanical origin of the propolis as leaves of an Asteraceae bush, *Baccharis dracunculifolia*[7].

Bioactive properties of this propolis type have been often attributed to artepillin C[8,10], and at present, the molecule has shown anti-microbial[8], anti-inflammatory[11], anti-oxidative[10], immuno-modulatory[12], and anti-carcinogenesis properties in humans[13,14]. Recently, a beneficial effect of artepillin C against obesity and diabetes was also reported as induction of brown-like adipocyte formation, when administrated to white adipocytes or even to mice orally[15]. This differentiation is mediated by direct binding of the compound to peroxisome proliferator-activated receptor (PPAR) γ, a master regulator of adipocyte differentiation, as an agonist[15]. Moreover, the prenyl moieties on the aromatic ring of artepillin C are key to the potency of its bioactivities[2,8,10,16].

Prenylation of aromatic rings in plants is catalyzed by prenyltransferases (PTs) belonging to the membrane-bound UbiA protein superfamily[17]. These enzymes cleave the diphosphate moieties from prenyl diphosphate substrates (prenyl donor substrates) using divalent cations as cofactors to generate prenyl carbocations, followed by transfer of these diphosphates to the aromatic substrates (prenyl acceptor substrates) of PTs through Friedel–Crafts alkylation[18]. Among the UbiA PT genes encoding enzymes dedicated to the biosynthesis of specialized phenolic classes are *Sophora flavescens* naringenin 8-dimethylallyltransferase-1 (*SfN8DT-1*), encoding a flavonoid-specific PT[19]; *Petroselinum crispum* PT1 (*PcPT1*), encoding a coumarin-specific PT[20]; *Hypericum calycinum* PT (*HcPT*), encoding a xanthone-specific PT[21]; *Humulus lupulus* PT-1 (*HlPT-1*), encoding a phloroglucinol PT[22], and a recently identified stilbenoid-specific *PT* genes[23,24]. In contrast, genes encoding PTs acting on simple phenylpropanes have

not yet been identified. Furthermore, assessments of multiple prenylated phenolic compounds have not well understood whether different PTs are responsible for individual prenylations, or whether one enzyme catalyzes the multiple prenylation reactions.

Biosynthetic genes to produce p-coumaric acid have been already revealed in plants[25], meaning that discovery of a PT gene accepting this phenylpropane molecule would enable synthetic biological production of artepillin C and its related compounds. This production system might improve the current situation of the limited availability of these molecules. Regardless of generally high contents of artepillin C along with its relatives in Brazilian green propolis, their contents strongly affected by various natural factors[26], being a serious issue of quality control in propolis production[5,6]. *B. dracunculifolia* can be considered as an alternative natural source, but the accessibility to this plant is strictly confined due to its major distribution within the southeastern parts of Latin America[7,27]. So far only chemical synthetic approaches have been reported to overcome these limitations in the availability of these compounds in nature[28,29].

The present study describes the identification of a phenylpropane-specific PT gene (*AcPT1*) that encodes an artepillin C synthase. The gene is isolated from an Asteraceaeous herbal plant, *Artemisia capillaris*, native to eastern Asia[30,31] using an RNA-seq library constructed from the leaves of this plant species by homology-based screening. This membrane-bound enzyme transfers two dimethylallyl moieties consecutively to p-coumaric acid to form artepillin C via drupanin, the mono-prenylated intermediate. To our knowledge, this unique multiple prenylation enzyme for phenolics in plants is the first phenylpropane-specific PT identified to date. Moreover, application of *AcPT1* to a metabolic engineering approach is used to construct a production system in yeast for artepillin C as well as for drupanin.

## Results

**Isolation of a phenylpropane-specific PT gene**. *A. capillaris* was selected as the source of the gene encoding phenylpropane-specific PT for two major reasons: (1) the Brazilian plant *B. dracunculifolia*, which accumulates large quantities of artepillin C, is not easily available in Japan, and (2) *A. capillaris* is an Asteraceaeous plant that produces capillartemisin A and B, which are hydroxylated derivatives of artepillin C[30], along with artepillin C, suggesting that PTs for phenylpropanoids are highly expressed. As this plant is native to Asian countries, it is easily available in Japan.

An EST library consisting of 39,615 unigenes was constructed from leaves of *A. capillaris* (Supplementary Fig. 1). These unigenes were subjected to a homology-based screening of candidate genes encoding phenylpropanoid-specific PTs. Aromatic substrate PTs involved in plant specialized metabolism share moderate amino acid identities (ca. 30–50%) with their probable ancestral UbiA PTs being involved in plant primary metabolism[19,21]. Hence, we searched for NCBI nucleotide collection for Asteracee polypeptide sequences in all six UbiA subfamilies involved in plant primary metabolism, i.e., homogentisate phytyltransferase (vitamin E 2-1, VTE2-1) for tocopherol, homogentisate solanesyltransferase (VTE2-2) for plastoquinone, 1,4-dihydroxy-2-naphthoic acid phytyltransferase (aberrant chloroplast development 4, ABC4) for phylloquinone, chlorophyll synthase (ATG4), protoheme IX farnesyltransferase (cytochrome *c* oxidase 10, COX10) for Heam *a*, and p-hydroxybenzoate polyprenyltransferase (PPT) for ubiquinone (Fig. 1)[17]. A tblastn analysis of our RNA-seq library using the six protein groups as queries yielded 18 hits, five of which showed high amino acid identities (>60%) with none of the queries

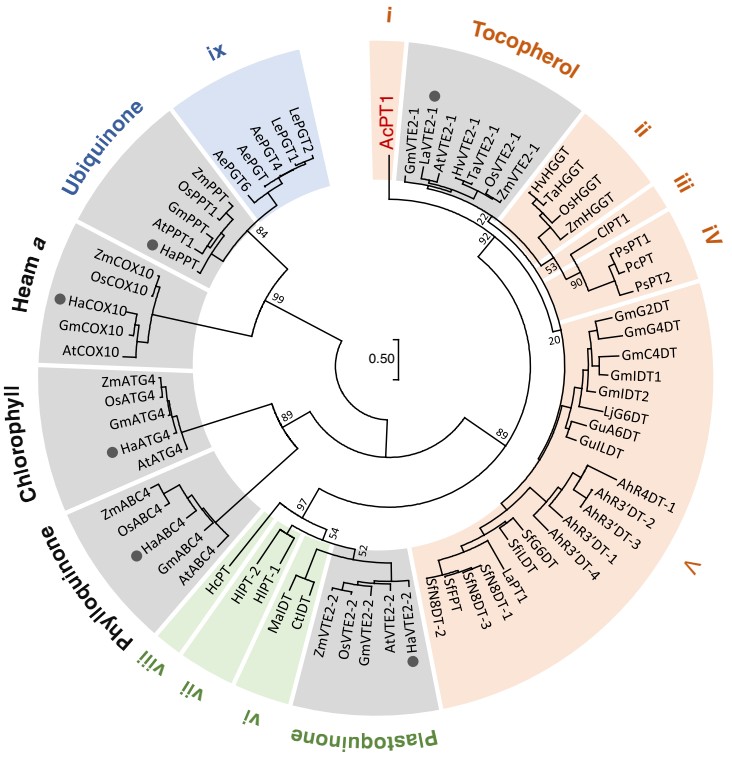

**Tocopherol-clade neighbors**

**i** : Asteraceae (phenylpropanes)

**ii** : Poaceae (homogentisic acid)

**iii** : Rutaceae (coumarins)

**iv** : Apiaceae (coumarins)

**v** : Fabaceae (flavonoids, stilbenoids)

**Plastoquinone-clade neighbors**

**vi** : Moraceae (flavonoids)

**vii** : Cannabaceae (phloroglucinols)

**viii** : Hypericaceae (xanthones)

**Ubiquinone-clade neighbors**

**ix** : Boraginaceae
(p-hydroxybenzoic acid)

**Fig. 1** Phylogenetic relationship of AcPT1 to the UbiA superfamily. A phylogenetic tree of members of the UbiA superfamily was constructed by the maximum likelihood method with 1000 bootstrap tests. Bootstrap values are shown for nodes separating clades. Clades related to primary metabolism are shown in gray, and each clade involved in specialized (secondary) metabolism was highlighted in color depending on the closest primary metabolism clade; i.e., orange, green, and blue for neighbors of tocopherol, plastoquinone, and ubiquinone clades, respectively. Gray circles represent Asteraceae PT polypeptides used as queries during in silico screening of AcPT1. The plant origin and accession number of each PT in the tree are described in Supplementary Tables 2 and 3

(Supplementary Fig. 2a). This result suggested that these five candidates differ functionally from all of the primary metabolite genes used as queries in *A. capillaris*. A subsequent Reads per kilobase of exon model per million mapped reads (RPKM)-based search for the five candidates showed that the expression of unigene 9515 was higher than that of the other genes (Supplementary Fig. 2b). The full coding sequence (CDS) of unigene 9515 was therefore isolated from a total RNA pool extracted from *A. capillaris* leaves by reverse transcription-polymerase chain reaction (RT-PCR)-based cloning. The resulting sequence, tentatively named *A. capillaris prenyltransferase1* (*AcPT1*), has been submitted to GenBank.

Using ChloroP and TMHMM software tools, *AcPT1* was found to encode a polypeptide of 401 amino acids with a predicted *N*-terminal transit peptide (TP) of 38 amino acids and multiple transmembrane regions, respectively (Supplementary Fig. 3). Both features are in conformity with other plant UbiA family PTs[19–21,32]. The AcPT1 amino acid sequence showed relatively high identity to VTE2-1, suggesting that the latter may be the

ancestral origin of AcPT1 (Supplementary Fig. 2a); nevertheless, it does not belong to any of the known clades of PTs (Fig. 1). This phylogenetic relationship suggested that this protein catalyzes novel prenylation reactions.

**In planta expression and subcellular localization of AcPT1.** Analysis of the organ-specific expression of *AcPT1* in intact *A. capillaris* plants by quantitative RT-PCR showed that *AcPT1* mRNA was present in leaves, stems, and flowers, with no clear statistically significant differences among them (Fig. 2a), although the expression in stems was higher than in other organs. Similarly, the content of artepillin C did not vary widely among the organs tested (Fig. 2b). *p*-Coumaric acid and drupanin, precursors of artepillin C were not detected in the specimens used in this study (Fig. 2b).

To investigate the subcellular localization of this protein in planta, the *N*-terminal 50 amino acids of AcPT1 were fused to synthetic green fluorescence protein (AcPT1TP-sGFP).

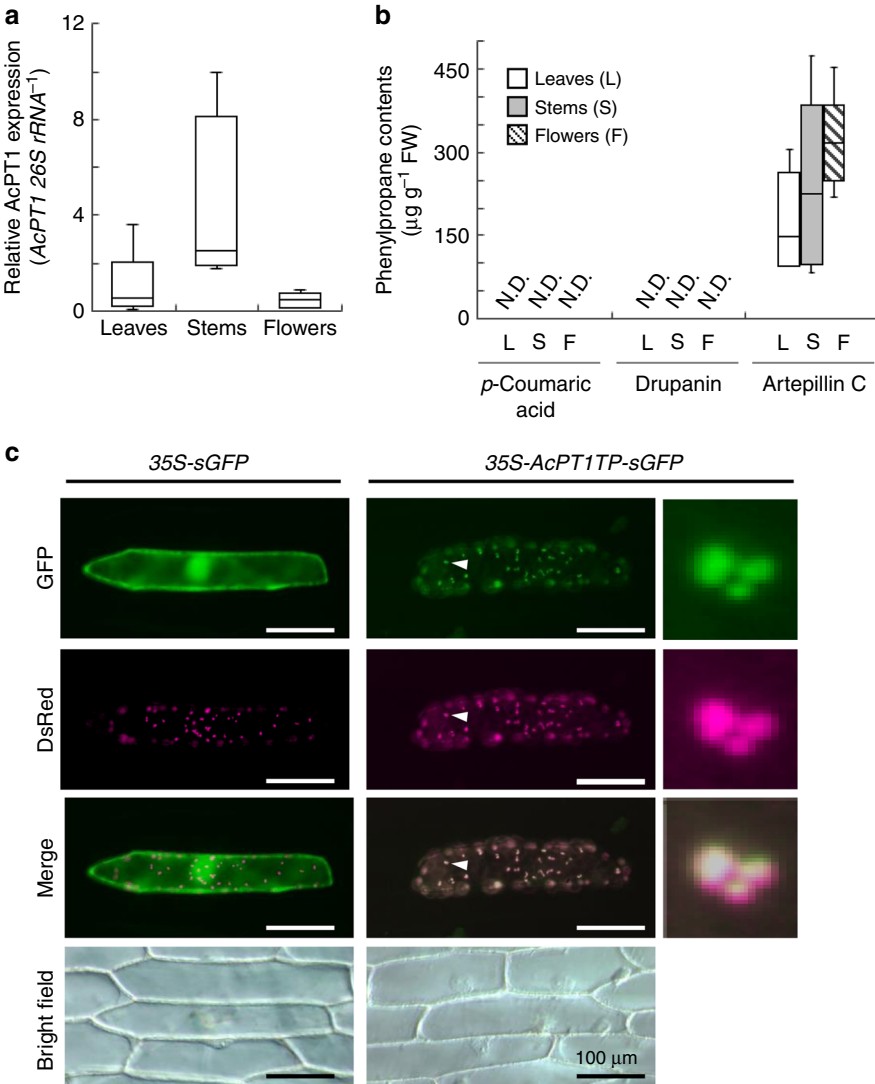

**Fig. 2** In planta expression and subcellular localization of AcPT1. **a**, **b** Gene expression levels of *AcPT1* relative to *Ac26S rRNA* (**a**) and the contents of three phenylpropane molecules, *p*-coumaric acid, drupanin, and artepillin C (µg g-fresh weight$^{-1}$) (**b**) in leaves, stems, and flowers of *A. capillaris*. The expression levels of *AcPT1* in aerial organs were normalized relative to *26S rRNA* and divided by the normalized expression level in leaves. Values ($n = 5$ biological replicates, except for stems ($n = 4$) in **a** are shown as box plots (center line, median; box limits, first and third quartiles; whiskers, minimum and maximum). In either **a** or **b**, clear significant differences between pairs were not observed by Tukey HSD test ($p > 0.05$). **c** Subcellular localization of AcPT1TP-sGFP in onion cells. Microscopic examination of onion cells co-expressing *sGFP* or *AcPT1TP-sGFP* together with *WxTP-DsRed*. As for AcPT1TP-sGFP, regions indicated by white arrows are enlarged. For merging, brightness and contrast of fluorescent images are unbiasedly adjusted with magenta used as a pseudo-color for the DsRed signal. Scale bars represent 100 µm

Microscopic analysis indicated that the sGFP fusion protein was distributed specifically to dotted structures in onion cells and almost completely matched the localization of the transit peptide region of waxy gene fused to *Discosoma* sp. red fluorescent protein (WxTP-DsRed)[33], a positive control for plastid localization (Fig. 2c). In addition, the AcPT1TP-sGFP protein was transiently expressed in *Nicotiana benthamiana* by agroinfiltration, which also showed that GFP fluorescence of AcPT1TP-sGFP colocalized with chlorophyll autofluorescence (Supplementary Fig. 4). These microscopic analyses strongly suggest that AcPT1 localizes to plastids in plant cells, in accordance with the localization of prenyl donor substrates for aromatic PTs, such as dimethylallyl diphosphate (DMAPP) and geranyl diphosphate (GPP), generated by the MEP pathway[32].

**p-Coumaric acid dimethylallyltransferase activity of AcPT1.**
AcPT1 was functionally analyzed using a transient expression system in *N. benthamiana*, because yeast expression system commonly used for membrane-bound enzymes often fails to express functional UbiA PTs. The full CDS of *AcPT1* was transiently expressed in *N. benthamiana* leaves by agroinfiltration, followed by the preparation of microsomes for in vitro enzymatic functional analysis. Incubation of these microsomes with *p*-coumaric acid and DMAPP in the presence of Mg$^{2+}$ yielded two enzymatic reaction products (P1 and P2) by high-performance liquid chromatography (HPLC) analysis (Fig. 3a and b). P1 and P2 were identified as drupanin and artepillin C, respectively, by direct comparisons of their retention times and tandem mass spectrometry (MS$^2$) spectra with standard specimens (Fig. 3b and Supplementary Fig. 5). Moreover, neither of these products was

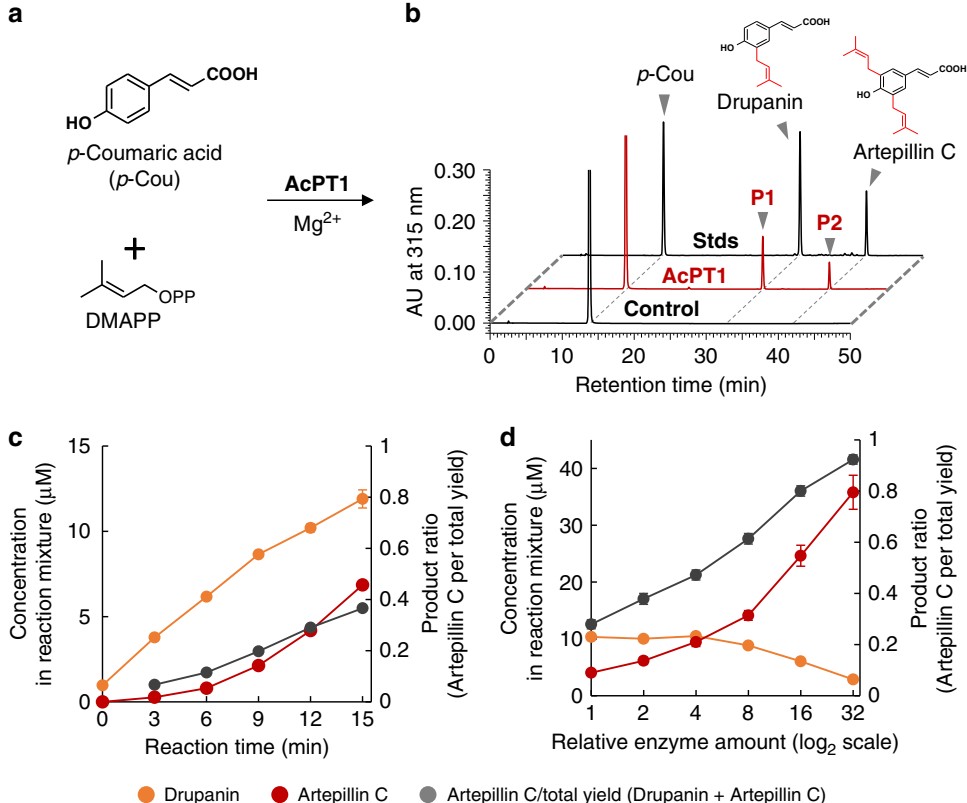

**Fig. 3** HPLC analysis of the *p*-coumaric acid:dimethylallyltransferase activity of recombinant AcPT1. **a** Components used in assays of *p*-coumaric acid dimethylallyltransferase activity of AcPT1. **b** HPLC chromatograms of enzyme reaction products. Microsomes prepared from *N. benthamiana* leaves expressing AcPT1 were used as crude enzymes. UV chromatograms at 315 nm of extracts from the reaction mixture are shown. The negative control consisted of microsomes prepared from *N. benthamiana* leaves expressing transit peptide-fused sGFP (AcPT1TP-sGFP). **c**, **d** Time (**c**) and enzyme-to-substrate (**d**) dependences of drupanin (orange circles) and artepillin C (red circles) synthesis and artepillin C per total yield (drupanin + artepillin C) ratios (gray circles). In both **c** and **d**, microsomes were incubated with 50 μM of *p*-coumaric acid and 0.50 mM of DMAPP. Values indicate means ± standard errors (*n* = 3 independent experiments)

detected in any of the negative controls (Fig. 3b and Supplementary Fig. 6a), indicating that recombinant AcPT1 catalyzes the double dimethylallylation of *p*-coumaric acid as a single polypeptide.

To further characterize these enzymatic reactions, the time-course of artepillin C synthesis was analyzed. Over an incubation time of 15 min, recombinant AcPT1 was able to produce both drupanin and artepillin C, with the molar ratio of artepillin C to the total yield of prenylated *p*-coumaric acids (artepillin C/(artepillin C + drupanin)) increasing from 7% at 3 min to 37% at 15 min (Fig. 3c). Later stages of these reactions were monitored by incubation with different concentrations of enzyme. As the amount of enzyme increased, the amount of artepillin C increased, while the amount of drupanin decreased (Fig. 3d), suggesting that drupanin is the substrate from which AcPT1 produces artepillin C. To confirm this finding, drupanin was tested as the direct substrate in the dimethylallyltransferase assay, resulting in the production of artepillin C (Fig. 4 and Supplementary Fig. 7a, b). Apparent $K_m$ values of AcPT1 for drupanin and DMAPP were calculated to be 50 ± 5 and 130 ± 20 μM (means ± standard errors), respectively (Supplementary Fig. 6b). These affinity constants are in the middle ranges for plant aromatic PTs, c.f., *Glycine max* (−)-glycinol 4-DT (GmG4DT), 68 μM for (−)-glycinol and 150 μM for DMAPP; SfN8DT-1, 55 μM for naringenin and 108 μM for DMAPP[19,32].

**Substrate specificity of AcPT1**. Recombinant AcPT1 accepts not only *p*-coumaric acid but drupanin as its substrate. To further

investigate its prenyl acceptor specificity in detail, other phenylpropane acids, with different substitution patterns on their aromatic rings, were tested (Fig. 4a). Ferulic acid was dimethylallylated by AcPT1, whereas the other compounds tested, such as cinnamic acid and *o*-/*m*-coumaric acids, were not (Fig. 4b and Supplementary Fig. 7c and d). Reduction of the carboxyl moiety at the propane chain terminal of ferulic acid to coniferyl aldehyde or coniferyl alcohol resulted in non-recognition of both by AcPT1 (Fig. 4a and b). In addition, shortening the propane chain of *p*-coumaric acid to *p*-hydroxybenzoic acid completely abolished AcPT1 activity (Fig. 4a and b). Other phenolic classes, such as umbelliferone and homogentisic acid, were also tested (Fig. 4a). Umbelliferone is a coumarin derivative, with a bicyclic backbone originating from phenylpropanes. Prenylated coumarin derivatives have been frequently isolated from Asteraceae[34,35], whereas homogentisic acid is the physiological prenyl acceptor for VTE2-1, the possible ancestor of AcPT1 (Fig. 1)[16]. Despite these relationships, neither of these phenolic compounds was converted to its prenylated form (Fig. 4b). Altogether, these in vitro experiments indicate that recombinant AcPT1 specifically accepts phenylpropane acids as its aromatic substrates by recognizing the hydroxyl moiety at the *p*-position on the aromatic ring, the carboxyl moiety at the propane chain terminal, and the length of the alkyl chain.

The substrate specificity of AcPT1 for prenyl donors was also assessed using prenyl diphosphates of different chain lengths, from C5 up to C15, in the presence of *p*-coumaric acid or drupanin as prenyl acceptor. DMAPP was the best prenyl donor

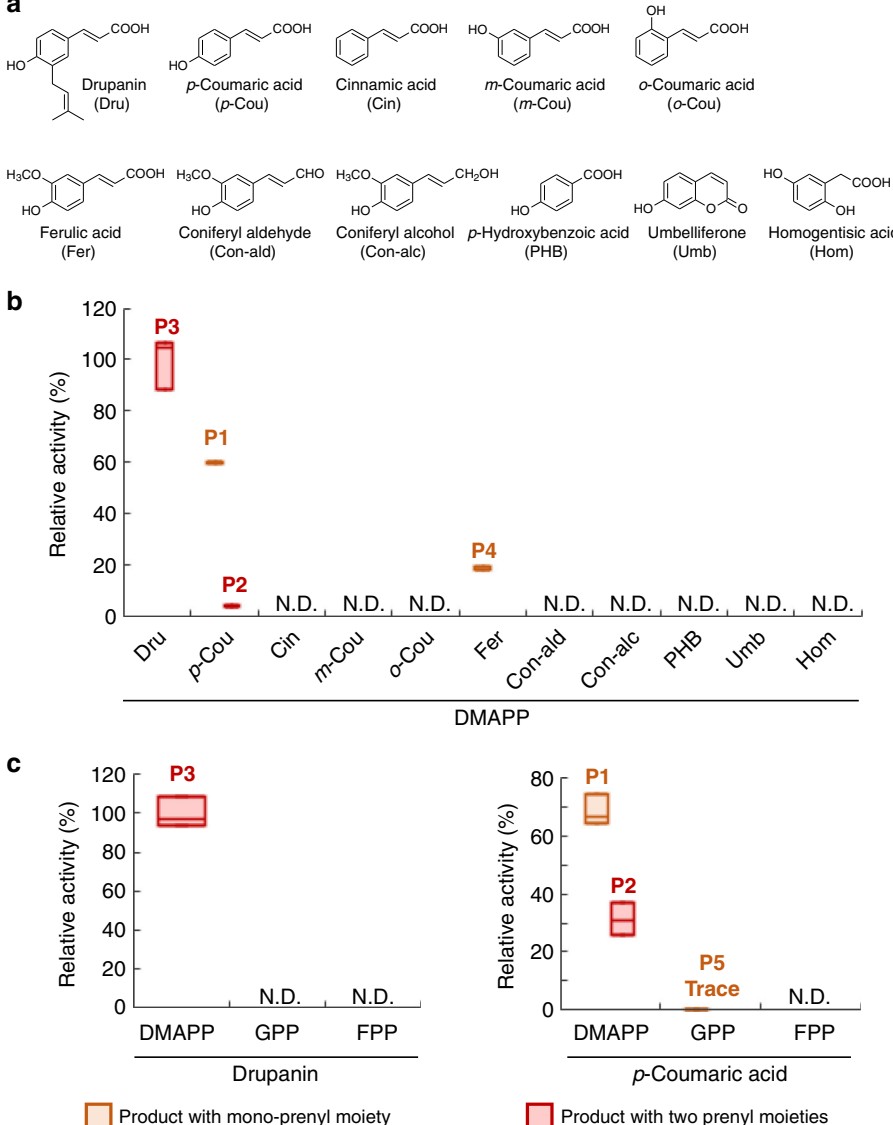

**Fig. 4 Substrate specificity of AcPT1. a** Chemical structures of prenyl acceptors tested. **b** Prenyl acceptor preference of AcPT1 in the dimethylallyltransferase reaction. **c** Prenyl donor preference of AcPT1 in using drupanin and *p*-coumaric acid as prenyl acceptors. The dimethylallyltransferase activity for drupanin (**b**, **c** for drupanin) or *p*-coumaric acid (**c** for *p*-coumaric acid) was set at 100%. Artepillin C (P2 and P3) and dimethylallylated ferulic acid (P4) were quantified as equivalents to drupanin and ferulic acid, respectively. Values (*n* = 3 independent experiments) are shown as box plots (center line, median; box limits, first and third quartiles or minimum and maximum)

substrate for both *p*-coumaric acid and drupanin, while GPP and farnesyl diphosphate (FPP) were almost completely unrecognized as prenyl donors for *p*-coumaric acid, except GPP yielded a trace amount of product (P5) (Fig. 4c). In most prenylation reactions for phenolics, the *ortho* position relative to the hydroxyl residue on the benzene ring is most often prenylated by UbiA PTs. This regio-specificity was strictly conserved during artepillin C synthesis by AcPT1 (Fig. 3). It is therefore likely that the reaction products resulting from the incubation of AcPT1 with ferulic acid (P4) and GPP (P5) follow this regio-specificity (Supplementary Fig. 7e, h).

**Production of prenylated *p*-coumaric acids in yeast.** Various bioactive plant metabolites have been recently synthesized in microorganisms using a synthetic biology approach, with a view toward their industrial production[36]. This study attempted to

utilize *AcPT1* to produce artepillin C in the budding yeast, *Saccharomyces cerevisiae*, as this host organism has a high capacity to express membrane-bound proteins[19,37,38].

A yeast strain (COUM11) has been reported to produce large amounts of *p*-coumaric acid following reconstruction of its biosynthetic pathway from phenylalanine using phenylalanine ammonia lyase (PAL), cinnamic acid 4-hydroxylase (C4H), and NADPH-cytochrome P450 reductase (CPR)[25]. *AcPT1* constructs were therefore introduced into COUM11, followed by its overexpression together with the overexpression of the gene set responsible for *p*-coumaric acid biosynthesis under the control of the galactose-inducible promoters, GAL1 and GAL10. However, neither drupanin nor artepillin C was detected in the yeast transformants, while a high level of *p*-coumaric acid was produced (Supplementary Fig. 8). The level of production of the prenyl acceptor molecule suggested that the level of DMAPP, the prenyl donor substrate, was too low in the yeast host.

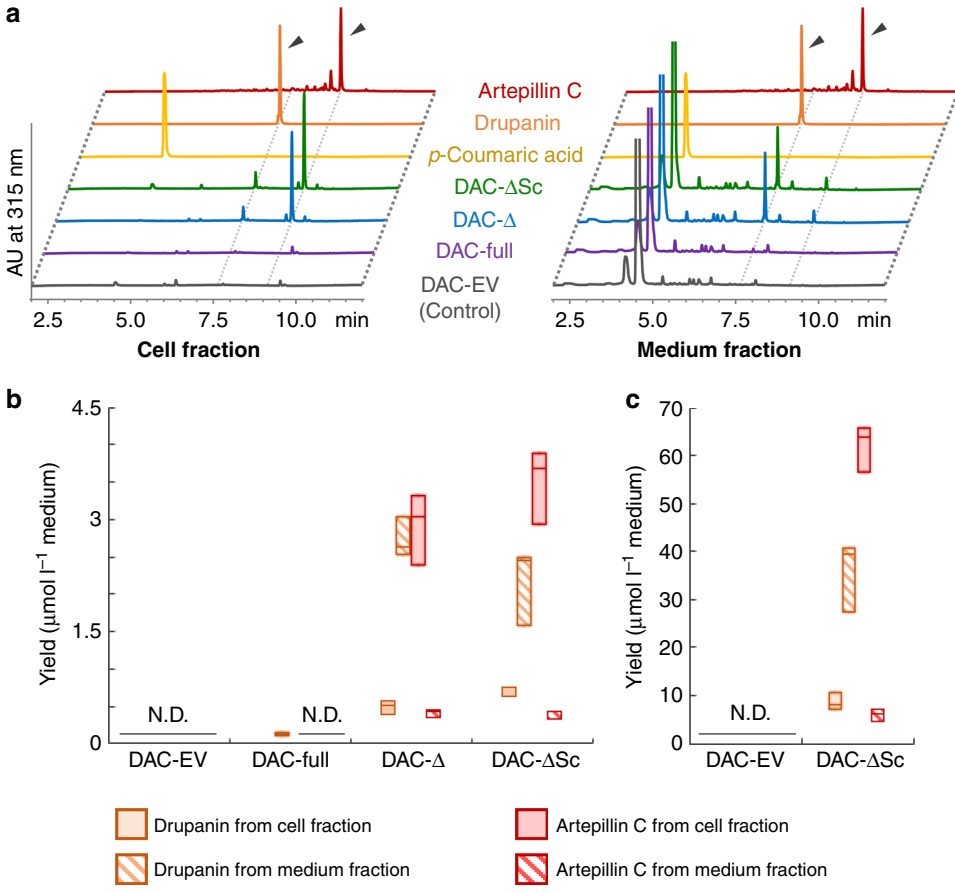

**Fig. 5** Drupanin and artepillin C production by DAC yeast strains. **a** UV chromatograms of cell extracts and the media of DAC strains. All chromatograms are shown on a comparative scale, except for those of standards. **b** Yields of drupanin and artepillin C in cell and medium fractions. Artepillin C was quantified as an equivalent to drupanin. **c** Effect of supplementation with 1.0 mM *p*-coumaric acid on the yields of drupanin and artepillin C. Values ($n = 3$ independent experiments) are shown as box plots (center line, median; box limits, first and third quartiles or minimum and maximum)

**Reinforcement of prenyl donor in vivo**. To increase the flux to DMAPP, the mevalonate pathway, which synthesizes DMAPP in yeast cells, was reinforced by the overexpression of *N*-terminal-truncated hydroxymethylglutaryl-CoA reductase (ΔNtHMGCoAR), an enzyme found to increase isoprenoid production up to several fold[37,39]. In addition to this upstream modification of the mevalonate pathway, an additional modification was introduced downstream. Specifically, mutations were introduced into endogenous *farnesyl pyrophosphate synthase* (*FPPS*) and *squalene synthase* genes[40], leading to a large DMAPP pool in yeast cells, with this mutant yeast strain, DD104, found to produce 44-fold more 8-dimethylallylated naringenin than the corresponding wild type[41]. Thus, *ΔNtHMGCoAR* and the gene set encoding *p*-coumaric acid production were introduced into the DD104 strain. The resulting yeast DD104 transformant (*ΔNtHMGCoAR/PAL/C4H/CPR*) was further transformed with pESC-His-*AcPT1* or pESC-His-empty vector (EV) as a negative control, strains designated as drupanin/ artepillin C production (DAC)-full or DAC-EV, respectively (Supplementary Fig. 9).

After propagation of DAC-full cells in liquid culture using minimal media containing glucose, the carbon source was replaced by galactose to induce expression of all foreign enzyme proteins. Twenty-four hours after induction, the culture was centrifuged, and phenolic compounds extracted individually from cells and medium. Drupanin but not artepillin C was detected in both extracts, especially from medium (Fig. 5a, b and Supplementary Fig. 10). These data suggest that the prenylation activity

of the strain was too weak to obtain sufficient drupanin substrate for the generation of artepillin C.

**Optimization of AcPT1 expression for yeast expression**. Transit peptides may negatively affect yeast production of plant PT proteins[32,38]. To overcome this problem, we constructed the DAC-Δ strain with *ΔTPAcPT1*, a truncated CDS missing its transit peptide-encoding region (Supplementary Fig. 9). As expected, this new transformant showed increased production of prenylated derivatives, 48-fold higher than the DAC-full strain, with equivalent amounts of artepillin C and drupanin produced (Fig. 5a and b). Codon-optimization of plant-derived biosynthetic genes for yeast expression shows reportedly two- to threefold increment in the production rate in yeast[42,43]. Therefore, we further modified DAC-Δ by codon optimization of *ΔTPAcPT1* for *S. cerevisiae* (*ΔTPAcPT1_Sc*) to create DAC-ΔSc (Supplementary Fig. 9), but this had little effect on total production of prenylated derivatives, with DAC-Δ and DAC-ΔSc producing 6.5 ± 0.5 and 6.8 ± 0.6 µmol l⁻¹ prenylated derivatives (means ± standard errors), respectively (Fig. 5a, b and Supplementary Fig. 10).

**Compartmentation of substrates and products**. To further enhance the production of artepillin C, we assessed the availability of both aromatic substrates (*p*-coumaric acid and drupanin) in greater detail. Although our fermentation system for the DAC-ΔSc strain contained a large quantity of *p*-coumaric acid, about 99%

was recovered from the medium, with only 1% collected from the cells (Fig. 5a and Supplementary Fig. 11a). Similarly, 75% of drupanin was detected in the medium of the DAC-ΔSc strain (Supplementary Fig. 11a), suggesting that the in vivo supply of the prenyl acceptors, especially *p*-coumaric acid, was limited. To validate this hypothesis, the DAC-ΔSc strain was incubated with galactose in the presence of an excess concentration (1.0 mM) of *p*-coumaric acid, or about 30-fold higher than the total yield of the three phenylpropanes ($32 \pm 2 \, \mu M$, means ± standard errors, $n =$ 3), to push the excreted *p*-coumaric acid back into the cells. Exogenous administration of substrate dramatically increased the production of both prenylated compounds by about 17-fold, to a total concentration of $113 \pm 7 \, \mu mol \, l^{-1}$ medium (means ± standard errors) (Fig. 5c). However, the ratio of the three phenyl-propanes was almost unchanged by this treatment regardless of its strong impact on total production rate; i.e., the aromatic substrates, particularly *p*-coumaric acid, were present mostly in the culture medium whereas artepillin C was present mostly in cells (Supplementary Figs. 11b and 12).

## Discussion

This study demonstrated the isolation of a gene from an Aster-aceaee plant *A. capillaris* encoding a phenylpropane-specific PT, AcPT1, and the in vitro functional characterization of this membrane-bound protein. This enzyme was shown to be highly specific to the *p*-hydroxyphenylpropane acid structure. PTs for other phenolic groups, including flavonoids, coumarins, phlor-oglucinols, stilbenes, and xanthones, have been described, while AcPT1 is the first phenylpropane-specific PT to be identi-fied to our knowledge. Moreover, this single polypeptide enzyme was shown to catalyze regio-specific successive prenylation reactions to yield multiple prenylated phenolic compounds. Among unigenes annotated as unknown function in our RNA-seq library, only *AcPT1* showed a high RPKM value. Subsequent real-time PCR validated the expression of this gene among dif-ferent aerial organs accumulating artepillin C. Furthermore, we also demonstrated the *p*-coumaric acid prenylation activity of AcPT1 in vivo in construction of the DAC yeast strains. These distinct facts strongly suggest that *AcPT1* contributes to the bio-synthesis of artepillin C in *A. capillaris*.

Phytochemical investigations have identified multi-prenylated phenolics in several plant taxa, including Cannabaceae, Fabaceae, Hypericaceae, and Rutaceae[1]. However, PT genes previously described encoded enzymes dedicated to transferring a single prenyl moiety in the biosyntheses of prenylated phenolic sub-stances even for phenolics with multiple prenyl chains[19,21,38]. A very recent example is di-prenylated xanthones produced by two independent PTs in *Hypericum* species (Hypericaceae)[44]. Char-acterization of these *Hypericum* PTs demonstrated that each of two PTs specifically catalyzes the first or the second prenylation, respectively, although they show trace activities for the other prenylation steps each other[44]. As the other example, di- and tri-prenylated phrologlucinol derivatives in hops (Cannabaceae) are synthesized by a functional metabolon composed of two inde-pendent PTs, although one of them shows only the first pre-nylation activity and the other is not active when it is alone[41]. In contrast to those enzymes, AcPT1 described in the present study consists of a single membrane-bound polypeptide that catalyzes sequential di-prenylation reactions with high efficiencies for each step. Another di-prenylation reaction is reported for *Morus alba* isoliquiritigenin DT (MaIDT), a flavonoid PT, which is capable of transfer of two dimethylallyl moieties to a xanthone molecule in vitro[45], but xanthone is not a physiological substrate of MalDT[45]. Outside plants, bacterial Mpz10 is a UbiA enzyme catalyzing a di-prenylation reaction in a marine *Actinomycete*,

while this enzyme shows low amino acid identity (<15%) to AcPT1[46].

Dissection of the di-prenylation reaction catalyzed by AcPT1 indicated that the two prenyl moieties are transferred individu-ally, being consistent with its capacity to bind to only one prenyl donor molecule. Similar to other UbiA PTs, AcPT1 possesses two aspartate-rich motifs, which were shown to cooperatively trap one prenyl diphosphate, a binding mediated by two divalent cation molecules as cofactors[18]. Intriguingly, we found that most of the drupanin produced by the DAC strains was recovered in the medium, indicating that AcPT1 releases this intermediate after the first prenylation reaction, but rarely directly reuses this molecule for the second prenylation reaction. This enzymatic feature may explain the presence of both mono- and di-prenylated phenylpropane derivatives in many Asteraceae plants, including *A. capillaris*[7,31,47]. However, our enzymatic assays showed that drupanin was the optimal prenyl acceptor for AcPT1. These seemingly contradictory properties suggest that synthesized drupanin is forced to leave the catalytic pocket, perhaps because AcPT1 may have to bind a prenyl donor prior to binding a prenyl acceptor. Alternatively, the aromatic substrate may move within the pocket and/or experience a change in angle during its condensation with the dimethylallyl carbocation gen-erated by DMAPP, preventing the product from returning to the initial state for the second prenylation. Another possibility is that the enzyme undergoes a conformational change during the reaction, which may promote the ejection of the product. To date, however, no other studies have assessed product release by members of the UbiA superfamily, suggesting the need for further investigations of the reaction mechanism, e.g., mutagenesis and crystallization, to determine the mechanism by which AcPT1 releases drupanin after its synthesis.

Another unique feature of AcPT1 is its clear preference for phenylpropanes as prenyl acceptor substrate. Phenylpropanes constitute a ubiquitous plant phenolic class with a C6-C3 chemical skeleton derived from phenylalanine and tyrosine[1]. This basic chemical structure, represented by a *p*-coumaroyl unit, serves as a universal aromatic building block in the con-struction of various classes of more chemically complex phe-nolic structures, such as flavonoids, coumarins, stilbenoids, and lignin polymers. These molecules allow plants to organize specialized phenolic assemblies in adapting to their environ-ments[35]. These species-specific adaptations resulted in the generation by the aromatic-core supplier itself of a wide variety of chemicals. For example, bioactive phenylpropane molecules with isoprene units are predominant in the Asteraceae and Rutaceae families[2,3]. Among these are artepillin C, the main product of AcPT1, and the active component of Brazilian green propolis[5].

The identification of *AcPT1* enabled us to try a synthetic biology approach to produce artepillin C in yeast. Modifications of the prenyl donor supply and removal of the transit peptide increased the total 24-h yield of drupanin and artepillin C to $6.8 \, \mu mol \, l^{-1}$ medium, exceeding the de novo production of 3-geranyl-4-hydroxybenzoate acid in engineered yeasts ($3.6 \, \mu mol \, l^{-1}$ medium following galactose induction for 48 h)[37]. We also realized that productivity of our strains can be improved. A clear barrier to be overcome is the poor accessibility of prenyl acceptors to the enzyme, due at least in part to the accumulation patterns of each phenylpropane molecule. Surprisingly, distribution was only slightly disturbed even by exogenous *p*-coumaric acid, which largely affected the yield. The strict localization patterns are see-mingly linked to the number of prenyl moieties. Alternatively, prenyl moieties are likely responsible for intracellular localization, which is also observed during yeast production of prenylated phloroglucinols and prenylated flavonoids[41,48]. Thus, this effect of

the prenyl chains can be extrapolated to a broad range of aromatic structures.

It has been reported that prenylated compounds remains in cells for a longer time than non-prenylated compounds[49]. This may be due to the greater hydrophobicity of prenylated derivatives, resulting in higher membrane permeability, and, more importantly, to the interactions between prenylated compounds and membrane proteins such as transporters[49]. Indeed, artepillin C has been found to permeate through Caco-2 cell monolayers mainly by passive diffusion[50]. In contrast, active transport systems largely contribute to the permeation of *p*-coumaric acid[51], suggesting that lipophilic chains affect the distributions of phenylpropanes produced in our yeast strains by a manner somewhat similar to that in mammalian cells.

The productivity of artepillin C in engineered yeast may be improved by localization engineering of molecules. As an attempt to overcome the low intracellular *p*-coumaric acid amounts, we investigated the *p*-coumaric acid accumulation capacity of yeast mutant lines lacking major efflux transporters, such as Δ *pleiotropic drug resistance 5* (Δ*pdr5*), Δ *sensitivity to 4-nitroquinoline-N-oxide 2* (Δ*snq2*), and AD12345678, in comparison with DD104 strain used in the synthetic biology[52,53]. Unexpectedly, however, the *p*-coumaric acid amounts retained in these mutant lines were rather smaller (50 ± 30–280 ± 60 nmol g$^{-1}$ cell, means ± standard errors, $n = 3$ each) than the control strain W303-1A (900 ± 200 nmol g$^{-1}$ cell, means ± standard errors, $n = 3$), the background of pdr5 and snq2 strains, and DD104 could only retain low level of *p*-coumaric acid (90 ± 40 nmol g$^{-1}$ cell, means ± standard errors, $n = 3$). Although a reasonable explanation of these results is not available now, a screening for a strain that retain a high level of *p*-coumaric acid will be a future research target to improve the productivity of artepillin C.

Alternatively, a phenylpropane importer may enhance the retention of endogenous substrate inside yeast cells, whereas overexpression of an efflux transporter for artepillin C may enhance the efficient production of this compound[54,55]. Productivity may also be enhanced by modifying the affinity constant, as AcPT1 has apparent $K_m$ values of 50 μM and 130 μM for drupanin and DMAPP, respectively, higher than that of other PTs. Alterations to the levels of other PTs, like *Citrus limon* PT1 (ClPT1) of lemon (6 μM for umbelliferone and 5 μM for GPP) may enhance prenylation velocity[38].

In conclusion, this study provides a novel example of divergent evolution of the UbiA superfamily in plant specialized metabolism, as shown by differences in prenyl acceptor preference and multi-prenylation mechanism. AcPT1 will be used for the mechanistic studies of prenylation reaction in future, and also for microbial production of artepillin C, where its narrow substrate specificity is beneficial to reduce byproducts and purification costs. The yeast production system shown here may contribute to the supply of a stable source of artepillin C as an alternative to propolis. Further chemical modifications of the core prenylated phenylpropane structure may also provide a variety of bioactive compounds in future.

## Materials and methods

**Plant materials and reagents**. *A. capillaris* plants for construction of RNA-seq were grown under natural conditions at the Tsukuba Division, Research Center for Medicinal Plant Resources, National Institutes of Biomedical Innovation, Health and Nutrition (NIBIOHN), Tsukuba, Japan (Plant Material No. 0014-06TS in NIBIOHN). *A. capillaris* plants for other experiments were maintained in the soil field of the Yamashina Botanical Research Institute, Nippon Shinyaku Co., Ltd. Standard specimens of drupanin and artepillin C were kindly provided by Api Co., Ltd. DMAPP, GPP, and FPP were kindly provided by Dr. Hirobumi Yamamoto (Toyo University), Dr. Tomohisa Kuzuyama (The University of Tokyo) and Dr. Takashi Kawasaki (Ritsumeikan University), and Dr. Seiji Takahashi (Tohoku University), respectively. Synthetic dextrose (SD) minimal media for yeast culture

were prepared essentially according to the protocol provided by Sigma-Aldrich (Catalog ID: Y2001).

**Construction of RNA-seq data from *A. capillaris* leaves**. Leaves of *A. capillaris* were harvested in August 2013 and frozen immediately in liquid nitrogen. Total RNA from frozen plant tissues was extracted using PureLink Plant RNA Reagent (Thermo Fisher Scientific), treated with recombinant DNase I (RNase-free) (TAKARA BIO), and purified using RNeasy Plant Mini Kits (Qiagen), according to the RNA clean-up protocol. A 10-μg aliquot of total RNA was used to construct a cDNA library using an Illumina TruSeq Prep Kit v2 according to the manufacturer's protocol (Illumina). The resulting cDNA library was sequenced using HiSeq 2000 (Illumina) with 100 bp paired-end reads in high output mode. Sequence reads were assembled de novo and their RPKM values calculated using CLC Genomics Workbench software ver. 5.5.2 (CLC Bio). A minimum contig length of 300 bp and the 'perform scaffolding' function provided assembled contigs after the removal of adaptor sequences and low-quality reads. After removing redundant sequences using the TIGR Gene Indices clustering tools (TGICL), 39,615 unigenes were assembled[56].

**In silico analysis**. Unigenes belonging to the UbiA superfamily were screened by tblastn using Bioedit software (http://www.mbio.ncsu.edu/BioEdit/bioedit.html), and their amino acid identities in comparison to queries were calculated. Plant UbiA proteins were multiply aligned by ClustalW, and a phylogenetic analysis was performed using a maximum likelihood method with 1000 bootstrap tests in MEGA6 software (http://www.megasoftware.net/). Transmembrane domains and the transit peptide sequence of AcPT1 were predicted with TMHMM Server v. 2.0 (http://www.cbs.dtu.dk/services/TMHMM/) and ChloroP (http://www.cbs.dtu.dk/services/ChloroP/), respectively.

**Quantification of phenylpropanes in *A. capillaris* organs**. Leaves, stems, and flowers of *A. capillaris* were frozen in liquid nitrogen and crushed to fine powder with mortars and pestles. Each sample was added to a tenfold volume of methanol, vigorously shaken at 2500 r.p.m. for 15 min and centrifuged at 10,000 × g for 5 min. Each supernatant was filtered with Minisart® RC4 (0.2-mm pore, Sartorius Stedim Biotech) and subjected to HPLC analysis. Phenolic compounds were quantified using a D-2000 Elite HPLC System equipped with a LiChrosphere RP-18 column (4 × 250 mm, Merck) and a L-2455 Photodiode array detector (Hitachi). Phenolic compounds in the extracts were separated with a gradient program of 20% (v/v) to 80% (v/v) solvent B (methanol with 0.3% (v/v) acetic acid) in solvent A (0.3% (v/v) acetic acid) for 55 min at 40 °C at a flow rate of 1 ml min$^{-1}$. Artepillin C was identified by comparison with a standard specimen, and quantified as an equivalent to drupanin because of the instability of standard artepillin C, which slowly decomposes over time.

**Isolation of *AcPT1* gene from *A. capillaris* leaves**. *A. capillaris* leaves were frozen in liquid N$_2$ and ground to fine powder with a mortar and a pestle, and total RNA was extracted using an RNeasy Plant Mini Kit (Qiagen), according to the manufacturer's instructions. Each total RNA sample was treated with DNase using a DNA-free™ Kit (Ambion), and first strand cDNA was synthesized by reverse-transcription with SuperScript™ III Reverse Transcriptase (Thermo Fisher Scientific) according to the manufacturer's protocols. Subsequently, the full CDS of *AcPT1* and its 5′-/3′-UTR regions was amplified by PCR using the cDNA as a template, TaKaRa LA Taq® DNA Polymerase (Takara), and the primer pair AcPT1_5′UTR_Fw and AcPT1_3′UTR_Rv1 (Supplementary Table 1). The amplicon was subsequently inserted into pGEM T-easy vector (Promega) by TA-cloning for sequencing.

**Real-time PCR for organ-specific expression of *AcPT1***. cDNA pools were prepared from the aerial parts (leaves, stems, and flowers) of *A. capillaris* plants, as described above, except that ReverTra Ace® qPCR RT Master Mix with gDNA Remover (Toyobo) was used for the RT reaction. Quantitative PCR analysis was performed using THUNDERBIRD® SYBR® qPCR Mix (Toyobo) and the primer pairs for AcPT1 (AcPT1_qRTPCR_Fw, Rv) and Ac26SrRNA (Ac26SrRNA_Fw, Rv) as a reference gene (Supplementary Table 1), controlled by CFX96 Deep Well (Bio-rad). The amplification program consisted of an initial denaturation at 95 °C for 2 min, followed by 50 cycles of denaturation at 95 °C for 15 s, annealing at 55 °C for 30 s, and elongation at 72 °C for 30 s.

**Expression of *AcPT1TP-sGFP* and microscopic analysis**. The 5′-terminal 150 bp of *AcPT1* (*AcPT1TP*) was amplified by PCR with KOD-plus-Neo (Toyobo) and the primer pair AcPT1_TP_Fw and AcPT1_TP_Rv (Supplementary Table 1), and the amplicon was inserted into the pGEM-T Easy vector (Promega) yielding the *AcPT1TP* sequence. This DNA fragment was inserted into the pENTR™/D-TOPO® vector (Invitrogen), followed by LR recombination with the pGWB505 vector to obtain the construct possessing *35Spro-AcPT1TP-sGFP*[57].

Particle bombardment of the plasmid into onion epidermal cells was performed with Biolistic® PDS-1000/He Particle Delivery System (Bio-Rad) and microscopic analysis was performed with VB-7000 Ver. 1.20 (Keyence)[19]. The plasmid

pHKN29 containing *CaMV35Spro-sGFP* and the plasmid pWxTP-DsRed were used as controls for free sGFP and plastid localization, respectively[33,58]. The AcPT1TP-sGFP protein was also expressed in epidermal cells of *N. benthamiana* leaves by agroinfiltration using LBA4404 strain[38]. Fluorescence images of transformed *N. benthamiana* were captured using FV3000 (Olympus, Japan) with a $20 \times 0.75$ numerical aperture objective. GFP was monitored by excitation at 488 nm with a 20-mW diode laser and emission at 500–540 nm, and chlorophyll autofluorescence was excited at 640 nm with a 40-mW diode laser and emission at 650–750 nm.

**Functional expression of *AcPT1* in *N. benthamiana* leaves.** The *AcPT1* CDS was amplified by PCR with KOD-Plus-Neo (Toyobo) using the primer pair AcPT1_-TOPO_Fw and AcPT1_TOPO_Rv (Supplementary Table 1), and the product was subcloned into the pENTR$^{\text{TM}}$/D-TOPO® vector by directional TOPO reaction, followed by LR recombination with the pGWB502 vector to yield a construct having 35Spro-AcPT1[57]. *AcPT1* was transiently expressed in *N. benthamiana* leaves by agroinfiltration and microsomes were prepared from the leaves using the pBIN61-P19 plasmid, as described[20,59].

**In vitro PT assay.** A reaction mixture (200 µl) containing 0.50 mM prenyl acceptor, 0.50 mM prenyl donor, 10 mM $MgCl_2$, and the microsome fraction was incubated at 30 °C for 60 min unless otherwise indicated. In a kinetic analysis, different concentrations of drupanin (15.6, 31.3, 62.5, 125, and 250 µM) and DMAPP (12.5, 25.0, 50.0, 100, and 200 µM) were incubated with the microsomes in the presence of 10 mM $MgCl_2$ and 0.5 mM the other substrate for 20 min. The enzyme reaction was terminated by adding 100 µl of 3 N HCl, followed by extraction with 300 µl of ethyl acetate. The organic phase was evaporated to dryness with $N_2$ gas and dissolved in 50 µl methanol. After centrifugation at $20,400 \times g$ for 5 min, the supernatant was subjected to HPLC and LC-ion trap (IT)-time of flight (TOF)/MS analyses for quantification and identification of enzymatic reaction products, respectively.

**LC-IT-TOF/MS analyses of enzymatic reaction products.** The reaction products were analyzed by HPLC as described above, except that drupanin dimethylallyl-transferase was assayed using an isocratic program of methanol:water:acetic acid (80:20:0.3). Reaction products were monitored at wavelengths of 280–330 nm, and prenylated phenolics were identified by LC-IT-TOF/MS (Shimadzu) in use of a TSK gel ODS-80Ts column ($2 \times 250$ nm, Tosoh) and a gradient program of 20% (v/v) to 80% (v/v) solvent B (acetonitrile with 0.3% (v/v) formic acid) in solvent A (0.3% (v/v) formic acid) for 60 min at 40 °C and a flow rate of 0.2 ml min$^{-1}$. Prenylated phenolics were identified using a positive ion mode with an *m/z* range of 50–500.

**Yeast platform for production of prenylated *p*-coumaric acids.** Total RNA was extracted from YPH499 yeast cells using RNeasy Mini Kit (Qiagen) and reverse transcribed with SuperScript™ III Reverse Transcriptase (Thermo Fisher Scientific). This cDNA pool was used as a template to amplify the *ΔNtHMGCoAR* sequence, which lacked the 1584-bp 5′-sequence, by PCR using the primer pair ΔNtHMGCoAR_Fw and ΔNtHMGCoAR_Rv (Supplementary Table 1). The amplified fragment was validated by sequencing in pGEM T-easy (Promega) vector. The plasmid was doubly digested with *Not*I and *Pac*I to obtain the *ΔNtHMGCoAR* sequence, which was introduced into multiple cloning site 1 of the pESC-Leu vector (Agilent Technologies), yielding the plasmid pESC-Leu-*ΔNtHMGCoAR*-CPR.

The pESC-Trp-*CPR* plasmid, possessing the *CPR* gene of hybrid poplar (*Populus trichocarpa × P. deltoidsand*) was kindly provided by Dr. Filippos Ververidis (Technological Educational Institute of Crete). Because the yeast host strain DD104 has Trp1 auxotrophy, the *CPR* CDS was PCR amplified using the primer pair (CPR_Fw and CPR_Rv (Supplementary Table 1). The amplicon was digested with *Bam*HI and *Xho*I and transferred into multiple cloning site 2 of the pESC-Leu-*ΔNtHMGCoAR* construct by In-Fusion® HD Cloning (Takara). The in-fusion product, pESC-Leu-*ΔNtHMGCoAR-CPR*, was subsequently introduced into the yeast strain DD104 harboring pESC-*hybrid poplar PAL-Glycine max C4H*[25].

To express *AcPT1* in *S. cerevisiae* cells, the codon of the truncated *AcPT1* lacking its 5′-terminal 150 bp (*ΔTPAcPT1*) was optimized by GeneArt™ Services (Thermo Fisher Scientific) to yield *ΔTPAcPT1_Sc*, which was inserted into the pMA-T vector (Thermo Fisher Scientific) for sequencing. The *ΔTPAcPT1_Sc* sequence was PCR amplified using the primer pair ΔTPAcPT1_Sc_EcoRI_Fw and ΔTPAcPT1_Sc_SacI_Rv (Supplementary Table 1), and the amplicon was cloned into the vector pCR™8/GW/TOPO® (Life Technologies). The TOPO reaction product was doubly digested with *Eco*RI and *Sac*I to yield the fragment *ΔTPAcPT1_Sc*, which was inserted into the multiple cloning site 1 region of the vector pESC-His (Agilent Technologies). The same procedure was used to construct the vectors pESC-His-*AcPT1* and pESC-His-*ΔTPAcPT1*, using the primer pair AcPT1_EcoRI_Fw and AcPT1_SacI_Rv (Supplementary Table 1) for *AcPT1* and the primer pair ΔTPAcPT1_EcoRI_Fw and AcPT1_SacI_Rv (Supplementary Table 1) for *ΔTPAcPT1*, respectively. Each construct was

introduced into DD104 harboring pESC-Leu-*ΔNtHMGCoAR-CPR* and pESC-Ura-*PAL-C4H*.

**Culture of engineered yeast cells.** Yeast colonies grown on agar plates of SD minimal medium lacking the amino acids his, leu, trp, and ura were cultured in 2 ml of SD (- his, leu, trp, ura) medium at 30 °C for 1 day at 200 r.p.m. A 20-µl aliquot of each preculture was inoculated into 10 ml of SD (- his, leu, trp, ura) medium and grown for an additional 2 days at 200 r.p.m. at 30 °C. The harvested yeast cells were washed once with 10 ml of sterile water and resuspended in 10 ml of synthetic galactose minimal medium under the same selection conditions as above. To apply exogenous *p*-coumaric acid, a stock solution (200 mM in ethanol) was added to the cultures to yield a final concentration of 1.0 mM just prior to galactose induction at 200 r.p.m. at 30 °C for 24 h. The cultures were centrifuged at $3000 \times g$ for 5 min; the cell samples were frozen at −80 °C if necessary, otherwise both cells and medium were extracted with ethyl acetate.

**Extraction of phenolic compounds from yeast cultures.** Each 10 ml aliquot of medium was mixed with 2 ml of 3 N HCl and 15 ml of ethyl acetate, followed by vortexing for 2 min and centrifugation at $3240 \times g$ for 10 min to separate aqueous and organic phases. Each collected ethyl acetate fraction was evaporated to dryness by vacuum centrifugation, and the precipitate dissolved in 250 µl of methanol. After centrifugation at $15,000 \times g$ for 5 min, the supernatant was subjected to HPLC and LC/MS analyses.

Yeast cells were washed with 1 ml of sterile water and resuspended in 5 ml of 100 mM potassium phosphate buffer (pH 7.0) for cell wall degradation by incubation with 100 units of Zymolyase™ 100-T (Zymo Research) at 200 r.p.m. at 30 °C for 30 min. The cell suspensions were disrupted by vortexing for 10 min in the presence of 3 ml of glass beads (diameter: 0.45–0.50 mm, Sartorius). After the addition of 1 ml of 3 N HCl and 7.5 ml of ethyl acetate, phenolic compounds were extracted by vortexing for 2 min.

**LC/MS analyses of yeast culture extracts.** Ultra performance LC (UPLC)-photodiode array (PDA) analysis was performed using a Shimadzu NEXERA UPLC equipped with a PDA detector (SPD M20A, from Shimadzu). Compounds were separated on a Phenomenex Kinetex XB-C18 column ($150 \times 2.1$ mm, 2.6 µm) heated at 40 °C. Mobile phases were composed of (A) water + 0.1% (v/v) formic acid and (B) methanol + 0.1% (v/v) formic acid, with flow rates of 0.3 ml min$^{-1}$. The gradient program consisted of 20% (B) for 0.74 min, 20–90% (B) over 7.26 min, 90–100% (B) over 2 min, 100% (B) for 2.5 min, and a return to the starting conditions in 0.01 min, followed by equilibration for 2.5 min. PDA cells were kept at 40 °C, and products were scanned at 190–450 nm.

HPLC/MS analysis was performed using an Ultimate 3000 system coupled to a PDA array detector and Linear Ion Trap Mass Spectrometer LTQ-XL (Thermo Electron Corporation). LTQ was equipped with an electrospray (ESI) interface operating in positive or negative ion mode. Xcalibur 2.1 software was used for computer control and data process. Compounds were separated on a Phenomenex Kinetex XB-C18 column ($150 \times 2.1$ mm, 2.6 µm) heated at 25 °C; with mobile phases consisting of (A) water + 0.1% (v/v) formic acid and (B) acetonitrile + 0.1% (v/v) formic acid at a flow rate of 0.3 ml min$^{-1}$. The gradient program consisted of 10% (B) for 1 min, 10–70% (B) over 11 min, 70–100% (B) over 2 min, 100% (B) for 2.5 min, and a return to the starting conditions in 0.5 min, followed by equilibration for 3 min. PDA cells were kept at 40 °C, and products were scanned at 190–800 nm. The sheath, auxiliary, and sweep gases were set at 40, 10, and 10 arbitrary units per minute, respectively, the capillary temperature at 300 °C, the spray voltage at 5 kV, the capillary voltage at 36 V, and the voltages of the tube and split at 80 and 44 kV for positive mode, respectively.

**Statistics and reproducibility.** $K_m$ values for DMAPP and drupanin were calculated by a non-linear least-squares method in Sigmaplot 12. All the other statistical analyses were performed by Tukey HSD test using R software version 3.4.1 (ref. [60]). *A. capillaris* organs (leaves, stems, and flowers) were collected from five different branches and used for both quantification of the expression level of *AcPT1* and the phenylpropanoid content. In vitro enzymatic assays were conducted with three independent incubations. The yield of prenylated *p*-coumaric acid derivatives by each yeast strain was quantified with three independent cultures.

**Reporting summary.** Further information on research design is available in the Nature Research Reporting Summary linked to this article.

## Data availability
Nucleotide sequences of AcPT1 (LC425153) and ΔTPAcPT1_Sc (LC425154) are available in NCBI. The raw RNA-Seq reads obtained in this study have been submitted to the DDBJ Sequence Read Archive (DRA) under accession number DRA007286.

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

## Acknowledgements

We thank H. Yamamoto (Toyo University) for providing DMAPP, Dr. T. Kuzuyama (The University of Tokyo) and Dr. T. Kawasaki (Kyoto University) for GPP, Dr. S. Takahashi (Tohoku University) for FPP, and Api. Co., Ltd. for standard specimens of drupanin and artepillin C. We are grateful to Dr. T. Nakagawa (Shimane University) for providing the pGWB vectors, Dr. T. Mitsui (Niigata University) for the pWxTP-DsRed plasmid, Dr. H. Kouchi (International Christian University) for the pHKN29 plasmid, and Dr. D. Baulcombe (University of Cambridge) for the pBIN61-P19 plasmid. The COUM11 strain and related plasmids (pESC-*PAL-C4H* and pESC-Trp-*CPR*) were kindly provided by Dr. F. Ververidis (Technological Educational Institute of Crete). Dr. M. J. C. Fischer (INRA, Colmar) and Dr. D. Werck-Reichhart (CNRS, University of Strasbourg) kindly provided the DD104 strain. Dr. T. Miyakawa (Hiroshima University) and Dr. A. Goffeau (Université Catholique de Louvain) also kindly provided Δ*pdr5* and Δ*snq2* strains, and AD12345678 strain, respectively. We also thank Mr. T. Hosouchi and Ms. S. Shinpo (Kazusa DNA Research Institute) for technical support with the Illumina sequencing. LC-IT-TOF/MS analyses were carried out with the Development and Assessment of Sustainable Humanosphere system of the Research Institute for Sustainable Humanosphere (RISH), Kyoto University. This work was supported in part by MEXT KAKENHI Grant No. 16H03282 (to K. Yazaki), by JSPS KAKENHI Grant No. 17H05441 (to K. Yazaki), by JSPS Overseas Research Fellowships (to R.M.), by "Bioprolor2" project (Région Grand-Est), by the "Impact Biomolecules" project of the "Lorraine Université d'Excellence" (Investissements d'avenir – ANR), by the New Energy and Industrial Technology Development Organization (NEDO, No. 16100890 to K. Yazaki), and by a research and development grant of the Japan Agency for Medical Research and Development (AMED; 18ak0101046h0003). An additional grant was also provided by the RISH, Kyoto University (Mission 5-1).

## Author contributions

R.M., A.S., and K. Yazaki conceived this work and designed the experiments. H. Suzuki, H. Seki, T.M., N. Kawano., K. Yoshimatsu, and N. Kawahara constructed the RNA-seq library. T.Y. provided the plant materials. R.M. performed in silico analysis. R.M. and T.T. quantified artepillin C in *A. capillaris* organs. R.M., T.T., A.S., and E.M. carried out the real-time PCR analysis. R.M., T.T., and K.T. analyzed subcellular localization. R.M. and T.T. characterized the enzymatic properties of AcPT1. R.M., T. T., and K. Yanagihara carried out the metabolic engineering of yeasts. R.M. and J.G. performed the UPLC and MS analyses of extracts of yeast cultures. F.B. and A.H. provided critical advice during the preparation of this manuscript. R.M. and T.T. analyzed the data, and R.M. and K. Yazaki wrote the manuscript with contribution from other authors.

## Competing interests

The authors declare no competing interests.
