## [Peer Review File · Communications Biology]

Editorial Note: This manuscript has been previously reviewed at another journal. This document only contains reviewer comments and rebuttal letters for versions considered at *Communications Biology*.

Reviewers' comments:

Reviewer #2 (Remarks to the Author):

The review comments are addressed well. Acceptance is suggested.

Reviewer #3 (Remarks to the Author):

Prenylation is one of the important structural modifications involved in the biosynthesis route of diverse secondary metabolites. So far, several plant aromatic prenyltransferase genes (flavonoid specific-, coumarin specific-, and phloroglucinol specific-) have been characterized from various plant species.

This study presented the identification of a phenylpropane-specific PT gene, AcPT1, from *Artemisia capillaris*. The membrane-bound AcPT1 could accept p-coumaric acid as its specific substrate and transfer two prenyl residues to yield a di-prenylated phenylpropane, artemipillin C, which was reported to possess various bioactivities. The authors also construct a production system in yeast for artemipillin C as well as for drupanin by metabolic engineering approach.

This work could provide useful hints to the investigation of plant prenyltransferases as well as the production of artemipillin C by engineering strains. However, the following points should be concerned more clearly before it could be published in *Communications Biology*.

1. This work is systematic and the research results are credible. However, the novelty of this enzyme was much limited. Although in the authors' response to Reviewer #3, the novelty of AcPT1 is defined as huge based on a sense of biosynthesis study, especially the author emphasized that AcPT1 can introduce two prenyl moieties to its physiological substrate. In my view, this point need more direct experimental evidence to support. The *in vivo* functional characterization results of AcPT1 should be added and deeply discussed to prove that p-coumaric acid was the native substrate of AcPT1.
2. The MS2 spectra (-18.0) of reaction product 5 in Supplementary Figure 7 did not show the characteristic fragmentation patterns of geranyl group. NMR data (at least high resolution mass spectrometry data) was needed to support the predicted chemical structure.
3. It was mentioned in the Discussion Section that knockout of an efflux transporter for p-coumaric acid in yeast may dramatically enhance the availability of *in vivo* substrates. This point may need more experimental evidence.

Minor point:

1. The potential use of ACPT1 for microbial production of artemipillin C was one of the significances mentioned in this work. To make it more clearly, it was recommended to introduce the available source of artemipillin C in the Introduction Section, such as the limitation or difficulties of its natural source and/or chemical synthesis.
2. The experimental procedure of kinetic studies should be more detailed in the methods section, since the kinetic parameter was a key factor to describe the property of an enzyme.

Response to Reviewers' comments

Reviewers' comments:

Reviewer #2 (Remarks to the Author):

The review comments are addressed well. Acceptance is suggested.

Thank you very much for your evaluation.

Reviewer #3 (Remarks to the Author):

Prenylation is one of the important structural modifications involved in the biosynthesis route of diverse secondary metabolites. So far, several plant aromatic prenyltransferase genes (flavonoid specific-, coumarin specific-, and phloroglucinol specific-) have been characterized from various plant species. This study presented the identification of a phenylpropane-specific PT gene, AcPT1, from *Artemisia capillaris*. The membrane-bound AcPT1 could accept p-coumaric acid as its specific substrate and transfer two prenyl residues to yield a di-prenylated phenylpropane, artemillin C, which was reported to possess various bioactivities. The authors also construct a production system in yeast for artemillin C as well as for drupanin by metabolic engineering approach. This work could provide useful hints to the investigation of plant prenyltransferases as well as the production of artemillin C by engineering strains. However, the following points should be concerned more clearly before it could be published in *Communications Biology*.

(Response) Thank you very much for the positive evaluations.

1. This work is systematic and the research results are credible. However, the novelty of this enzyme was much limited. Although in the authors' response to Reviewer #3, the novelty of AcPT1 is defined as huge based on a sense of biosynthesis study, especially the author emphasized that AcPT1 can introduce two prenyl moieties to its physiological substrate. In my view, this point need more direct experimental evidence to support. The in vivo functional characterization results of AcPT1 should be added and deeply discussed to prove that p-coumaric acid was the native substrate of AcPT1.

(Responses) Thank you very much for the valuable comments. We conducted additional experiments on in vivo function of AcPT1 according to the comments. As *A. capillaris* is a non-model plant whose transformation method has not been established, we used *N. benthamiana* transient expression system, which is often used to characterize *in vivo* function of proteins derived from such plant species, including PcPT1 that is a UbiA PT from parsley and prenylate umbelliferone (Karamat et al., Plant J, 2014). Therefore, we tried *in vivo* prenylation of *p*-coumaric acid in *N. benthamiana* cells expressing AcPT1. Finally, however, we could not detect the synthesis of prenylated *p*-coumaric acid derivatives in spite of several modifications as follows.

- 1) First, we introduced the full CDS of *AcPT1* in *N. benthamiana* by agroinfiltration, and four days later, 1 mM of *p*-coumaric acid solution (0.5% EtOH) was infiltrated into leaves expressing *AcPT1* and incubated them further for 1 day with their petioles soaked in the same 1 mM *p*-coumaric acid solution. Phenolic compounds were extracted from the leaves with methanol and analyzed by UPLC/MS. However, we could not observe the synthesis of drupanin, artemillin C or other possible prenylated derivatives.

There are several reasons for this. *p*-Coumaric acid has a carboxyl moiety. Hence, after entering into cells, the carboxy residue of *p*-coumaric acid takes the anionic form in the cytosolic pH at 7.2 – 7.5. The ionic form has usually a low membrane permeability, and thus this substrate could hardly go across the chloroplast envelopes, while AcPT1 is localized in chloroplasts. Then, we assume that the accessibility of the phenolic substrate to AcPT1 is too low to produce enough amounts of prenylated *p*-coumaric acids for the detection by PDA or MS. In addition, in *N. benthamiana* as well as many other plant species, *p*-coumaric acid is a good substrate for many endogenous metabolites such as chlorogenic acid derivatives that are common in *Nicotiana* spp., flavonoids and lignin. It is good possible that *p*-coumaric acid is rapidly metabolized by endogenous enzymes before its access to AcPT1.

- 2) As the second attempt, the subcellular localization of AcPT1 was modified from plastids to cytosol area. Adding both the ER-localizing signal peptide of a lectin from the common bean and the ER-retention signal sequence

(KDEL) to the N- and C-termini of AcPT1 proteins, respectively, we tried to redirect the membrane protein AcPT1 to ER, according to references (Ohara et al., 2003; Ohara et al., 2004). We expressed the fusion protein (ERsignal- Δ TPAcPT1-KDEL) in parallel with the simple truncated version of AcPT1 (Δ TPAcPT1), and tested *in vivo* prenylation by these engineered AcPT1 derivatives in a similar method as applied to the full length AcPT1. However, we again failed to detect the prenylated *p*-coumaric acids in these transient transformants, either ERsignal- Δ TPAcPT1-KDEL or Δ TPAcPT1.

We also prepared microsomes from *N. benthamiana* leaves expressing Δ TPAcPT1 or ERsiglectin- Δ TPAcPT1-KDEL, and measures the *p*-coumaric acid prenylation activity of these microsomes *in vitro*, resulting in approximately 25 and 2000-fold less activities than microsomes for the full-length AcPT1. This is probably because of the enforced alteration of subcellular localization of the membrane enzyme.

Taken together, a huge work will be needed to properly characterize *in vivo* function of AcPT1 in plant cells at this moment. Because the subcellular localization of this enzyme could not be changed in keeping its catalytic activity in this study, a possible solution would be the identification of a transporter that mediates the import of *p*-coumaric acid from cytosol into plastids at the plastid envelopes. The existence of such transporters have been implied in the PT studies involved in prenylated flavonoids and tocopherol/plastoquinone biosynthesis (A. Sugiyama et al., Met Eng. 2011; MS Laurent, Antioxidants, 2018), while no transporter molecule is identified thus far. New discovery of such transporter is far beyond the present manuscript.

Because we could not characterize AcPT1 function in living plant cells, we have modified the discussion on its *in vivo* function in *A. capillaris* considering the consistency of its enzymatic function between DAC yeast cells and *N. benthamiana* expressing *AcPT1* (line 291-303). We hope that Reviewer#3 understand the difficulty in *in planta* characterization of AcPT1 and our modification in this discussion part is convincing.

2. The MS2 spectra (-18.0) of reaction product 5 in Supplementary Figure 7 did not show the characteristic fragmentation patterns of geranyl group. NMR data

(at least high resolution mass spectrometry data) was needed to support the predicted chemical structure.

The conversion rate of AcPT1 to yield the product 5 is too low in this assay system to collect enough amounts for NMR analysis. Instead, we replaced MS and MS² spectra of the product using those with a higher resolution (Supplementary Figure 7g). Additionally, those of the product 3 (dimethylallylated ferulic acid) was shown in Supplementary Figure 7d, as this compound was also assessed without standards similarly to the product 5.

3. It was mentioned in the Discussion Section that knockout of an efflux transporter for p-coumaric acid in yeast may dramatically enhance the availability of in vivo substrates. This point may need more experimental evidence.

Budding yeast has a powerful drug efflux pump with broad specificity, an ABC transporter named PDR5, which recognizes more than a hundred of different compounds. Another ABC transporter, SNQ2 is also known to be a broad range drug efflux transporter. We first presumed that either one may be responsible for excrete p-coumaric acid from the yeast cells. To evaluate this presumption, we conducted an assay, in which several yeast strains lacking drug efflux pumps were incubated with 1 mM p-coumaric acid and the remaining amount in the cells was quantitated by HPLC. Surprising, however, the cellular content of p-coumaric acid strongly varies, even unexpectedly *pdr5* (151 ± 26.5 nmol/g) and *snq2* (278 ± 62.9 nmol/g) gave lower content than the control strain W3031A (853 ± 240 nmol/g), whereas AD12345678 (51.2 ± 27.7 nmol/g) and DD104 (94.8 ± 37.6 nmol/g) used in this study gave also apparently low retainment of p-coumaric acid. We repeated these experiments at least twice. These results suggest that a thorough screening is necessary to increase the subcellular availability of p-coumaric acid, and we have to start the metabolic engineering in the candidate strain from the beginning. This takes too long time and we would like to do this as a future research topic. Reflecting the result, we have explained the strong fluctuation among yeast strains in retaining p-coumaric acid inside the cells, and a screening will be effective to increase the substrate availability, in the revised manuscript (line 376–386).

Minor point:

1. The potential use of ACPT1 for microbial production of artemisinin C was one of

the significances mentioned in this work. To make it more clearly, it was recommended to introduce the available source of artemisin C in the Introduction Section, such as the limitation or difficulties of its natural source and/or chemical synthesis.

Thank you for the valuable advice. We have added a paragraph explaining the current situation of the limited availability of artemisin C from nature and previous chemical synthesis approaches to overcome it in Introduction, line 90-100 of the revised manuscript.

2. The experimental procedure of kinetic studies should be more detailed in the methods section, since the kinetic parameter was a key factor to describe the property of an enzyme.

We have described concentrations of substrates and the incubation time about our kinetic analysis in line 505-507 in Materials and Methods of the revised manuscript.

REVIEWERS' COMMENTS:

Reviewer #2 (Remarks to the Author):

All the issues are well addressed, and the present manuscript is acceptable. Congratulations!